# Investigation of the Performance of a Hybrid Dryer Designed for the Food Industry

**DOI:** 10.3390/foods8020081

**Published:** 2019-02-21

**Authors:** Soner Çelen, Mehmet Akif Karataser

**Affiliations:** 1Department of Mechanical Engineering, Namık Kemal University, Tekirdağ 59860, Turkey; 2Vocational School of Technical Sciences, Namık Kemal University, Tekirdağ 59030, Turkey; makarataser@nku.edu.tr

**Keywords:** black radish, conveyor, microwave drying, modeling, solar energy

## Abstract

In this study, a microwave energy conveyor dryer with the support of solar energy was researched with respect to its usability in the food industry and its short drying time and low energy requirements. The hot air produced using a collector (which was designed in the shape of a hemisphere for the purpose of getting high efficiency from solar energy) was transferred to a conveyor drying chamber operating with microwave energy at the speed of 0.245 m/min. Drying kinetics of the selected black radishes for the experiments were determined using samples sliced into 4, 6, and 8 mm. At the end of the drying, the duration and specific energy consumption for the drying process as well as color changes in the samples were measured, and the collector’s efficiency was calculated. In addition, as a result of statistical analysis, the most suitable model among the seven drying models was determined.

## 1. Introduction

The drying process is the removal of water or other liquids from gases, liquids, or solids. Most commonly, the term “drying” refers to the removal of water or other volatile substances from additives by thermal methods [1]. The aim of drying is to reduce the volume and the weight to facilitate transportation, to sterilize the products, and to produce materials that provide the desired conditions. 

In terms of energy saving, the major percentage of energy consumption in many industries is formed by the energy consumed for drying processes. With the usage of new technologies it is possible to save a considerable amount of energy. The main factor to be taken into consideration when selecting a drying process and a dryer is to use minimum energy and a maximum drying rate to obtain the product with the desired qualities [2]. 

Microwaves are electromagnetic waves in the frequency range of 300 MHz to 300 GHz [3]. The heating effect of the microwave can be explained as the ingestion of energy by a dielectric material and then an increase in the temperature as the result of the absorption of this material. There are two important mechanisms—Ionic polarization and dipole rotation—That explain the heat consumption in the microwave field [4]. Electromagnetic energy in the systems of microwave heating is transformed into heat by the water molecules vibrating millions of times inside the direct substance. This vibration and the resulting energy ensure that the dampness in the material evaporates very rapidly [5]. Therefore, the heat is transferred from the inside of the material to the outside, as opposed to conventional heating systems in microwave heating. In microwave drying, since it directly penetrates the heat and the material, the process is much faster and the duration is shorter than in conventional drying systems [6]. Microwave drying is an alternative drying method that provides advantages such as high heat conduction to the inner parts of the dried material, cleaning, energy gain, easy process control, and a quick start and termination of the drying process [7]. In this method, dielectric properties, mass of the material, specific heat, geometry, heat dissipation mechanism, power produced in the material, and the efficiency of the power produced in the microwave system are taken into consideration as the basic parameters [5]. However, when used alone, the microwave drying method causes negative effects such as irregular heat distribution on the products, textural damage, high investment costs, and limited effect of the microwave rays on the products [8]. In order to reduce these problems and provide faster and more efficient drying processes, it might be appropriate to consider the application of microwave and other drying methods together in the drying process of the food.

In order to design a suitable microwave dryer, drying kinetics data from experimental studies are necessary. However, it is impossible to investigate all the drying conditions just by experiments due to time and high cost. Therefore, drying models are developed for various of materials [9]. Many studies have been carried out with these models on drying foods such as carrageenan [10], cocoa [11], apple and red bell paper [12], plum [13], rice [14], prina (crude olive cake) [15], carrot [16], parsley leaves [17], organic tomato [18], and mushroom [19]. In addition, the microwave-convective drying of food has been researched by many researchers in recent years. When the studies on drying included in the literature were examined, the process of drying was carried out by using microwave-vacuum [20,21,22], microwave-hot air combination dryers [23,24,25,26], and microwave dryers [27].

Air solar-collectors, also called solar-air heaters, are simple devices which are durable, lightweight, and suitable for volume heating without frost and corrosion problems. Solar air collectors can be easily used together with additional heaters in building heating and drying of the agricultural products. In addition, in medium- and low-temperature applications, the required hot air can be produced with solar air heaters.

Solar energy is one of the widely used renewable energy sources in drying. Drying of vegetables and fruits using renewable energy such as solar drying has less environmental impact. It is one of the oldest applications of solar energy to preserve food. The usage of solar energy in the drying is becoming an important and viable alternative. In this research, the black radish drying process was carried out using solar and microwave energy. Microwave energy was selected because the products can be brought to desired temperature levels in a shorter time compared to the conventional methods due to volumetric heating [28]. For this reason, microwave technology provides faster heating than conventional methods, and the energy requirement is low while energy efficiency is high. Also, it is used as an alternative to conventional methods as it involves easy cleaning of equipment, occupies less space, is applicable to packaged foods, and preserves nutritional value [29]. This design has been realized by taking these advantages into consideration. The main purpose of this research is to investigate the performance of the newly designed combined dryer in the food sector with low energy. In addition to this, another purpose is to compare the moisture content values obtained experimentally from the dried products with the estimated moisture content values using the seven different thin-layer drying methods used in this study to identify the model that best describes the experimental data.

## 2. Materials and Methods

### 2.1. Food Material and Drying System

A black radish (*Raphanus sativus* L. var. *Niger*) with approximately 85% ± 0.5% (wet base) (measured by drying at 105 °C in 24 h) moisture content was provided from a local market stored at 4 °C located in Tekirdağ City. The black radishes were peeled and sliced at 4 ± 0.3 mm, 6 ± 0.2 mm and 8 ± 0.2 mm thicknesses by the knife for the experiments. Different samples were used for each experiment.

In the design and manufacture of the solar collector, basic parameters such as economic efficiency, easy accessibility of the collector’s materials, manufacturing possibilities, and thermal properties are taken into consideration. The experiment setup consists of a solar collector, a circulation fan, a drying chamber, two magnetrons, and one conveyor (Figure 1) [30]. 

It was made of acrylic material with a high thermal permeability and was designed as a sphere with a diameter of 1.5 m and a height of 0.6 m to take the sunlight straight from every possible directions. Thus, the heat of the air in the collector reached a high temperature in a short time and a short drying period was achieved, which is important in the food drying industry. In order to provide air circulation, the collector was placed at the stands with a height of 0.5 m and air inlet channels were opened to the bottom surface of the collector. The exhaust sheet metal placed on the bottom surface of the collector was painted with black enamel (α = 0.97) (ε = 0.97). In order to send warm air to the drying chamber, the carrier aluminum cylindrical channel (1 m × Ø0.2 m) was placed on the top surface of the collector where the temperature is the highest. Probes were used to measure the center and upper temperature values of the collector. The bottom part of the collector and the surrounding of the carrier channel are insulated to prevent heat losses. The intensity of the radiation on top of the collector was measured and recorded with a CEM DT-185 brand pyrometer during the drying process. A speed-adjustable fan (HCM-180 N, İstanbul, Turkey) with 35-W power was used inside the duct. The other end of the duct was connected to the drying chamber. An anemometer was connected to the duct outlet to measure the temperature probe and hot air velocity entering the drying chamber. The measured values were read out on the device and recorded. The drying chamber (3.5 × 0.4 × 0.4 m) is a conveyor system consisting of a Teflon belt. Two magnetrons which transmit microwave energy are connected to the inlet and outlet ends of the drying chamber. The setting of the microwave energy and conveyor speed was made from the control panel. The energy consumed during drying process was measured by an energy-measuring device connected to the system. In addition, a PT100 temperature gauge was used to measure the temperature in the drying chamber.

### 2.2. Experimental Procedure

The experiments were held between 15 and 17 August in Tekirdağ City, Turkey. No pretreatment except peeling and slicing was applied to the radish before the drying experiments. After the slicing process, color parameters were measured with a colorimeter for fresh radish sample. Drying experiments were carried out with microwave power values of 0.7 kW, 1 kW, and 1.4 kW (at 2450 MHz) with a belt speed of 0.245 m/min. Weight loss values of the samples during drying were measured with the Presica XB 620 M (Precisa Instruments AG, Dietikon, Switzerland) brand precision scale of accuracy ± 0.001 g every 5 min. To determine the initial moisture value, the dry weight of the material was determined at 105 °C for 24 h. Initial sample weights ranged from 11 g to 13 g, depending on the slice thickness. The experiments were carried out according to the weather conditions. An outdoor environment where the sun was effective without shade was preferred. Prior to drying, conveyor speed and microwave power level were set on the control panel. At 11:30 h, when the air in the collector exposed to the sun reached to a constant temperature, the drying process was started. The microwave energy worked continuously throughout the drying during the experiments. Drying process lasted until 12% ± 0.5% (wet base) moisture level was reached. After drying, color parameters and energy consumption were measured and recorded. Each experiment was repeated three times in order to get the correct result.

### 2.3. Modeling of Drying

Different products and systems should be analyzed with different models which were created by using heat, mass and momentum transfer and material knowledge. Effectively, modeling the drying behavior is important for investigation of drying characteristics of samples. In this study, with the help of the nonlinear regression analysis method, the modeling of the moisture rate time variation curves obtained from the experimental results at different microwave power values was performed. The moisture ratio is defined as given in Equation (1).
MR = (m − m_e_)/(m_0_ − m_e_)(1)

The value of equilibrium moisture content (m_e_) is relatively small compared to m, which represents the wet basis moisture content at any time, and initial moisture content (m_0_), where the error implied in the simplification is negligible [31]. Therefore, the equilibrium moisture content was assumed to be zero for microwave drying. Drying curves were obtained for the black radish. The experimental moisture ratio versus drying time data were fitted in seven thin-layer drying models, as shown in Table 1.

In order to determine the most appropriate model, some statistical parameters need to be calculated. These are the *r*^2^, *e*_s_, and *χ*^2^ values. The correlation coefficient (*r*^2^) is one of the criteria used in determining the curve fit. In addition to the correlation coefficient, the estimated standard error (*e*_s_) and chi-squared (*χ*^2^) are the other parameters used in determining of the efficiency degree of regression fitness [32]. To get the best fit of the experimental data, the coefficient of determination should be higher and the χ^2^ and *e*_s_ values should be lower. These parameters are defined as in Equations (2)–(4) [19].
(2)$$r=\frac{n_{0}\sum_{i=1}^{n_{0}}{\mathrm{MR}}_{teo,i}{\mathrm{MR}}_{exp,i}-\sum_{i=1}^{n_{0}}{\mathrm{MR}}_{teo,i}\sum_{i=1}^{n_{0}}{\mathrm{MR}}_{exp,i}}{\sqrt{n_{0}\sum_{i=1}^{n_{0}}{\left({\mathrm{MR}}_{teo,i}\right)}^{2}-{\left(\sum_{i=1}^{n_{0}}{\mathrm{MR}}_{teo,i}\right)}^{2}}\sqrt{n_{0}\sum_{i=1}^{n_{0}}{\left({\mathrm{MR}}_{exp,i}\right)}^{2}-{\left(\sum_{i=1}^{n_{0}}{\mathrm{MR}}_{exp,i}\right)}^{2}}}$$
(3)$$e_{s}=\sqrt{\frac{\sum_{i=1}^{n_{0}}{\left({\mathrm{MR}}_{teo,i}-{\mathrm{MR}}_{exp,i}\right)}^{2}}{n_{0}}}$$
(4)$$\chi^{2}=\frac{\sum_{i=1}^{n_{0}}{\left({\mathrm{MR}}_{teo,i}-{\mathrm{MR}}_{exp,i}\right)}^{2}}{n_{0}-n_{c}}$$
where MR*_teo,i_* is the *i*th predicted moisture ratio, MR*_exp,i_* is the *i*th experimental moisture ratio, *n*_0_ is the number of observations, and *n*_c_ is the number of coefficients in the drying model.

### 2.4. Color Measurement

Color measurements for fresh and dried black radish samples were made using a colorimeter. Used parameters are L (brightness), a (redness/greenness), b (yellowness/blueness), and C (chroma) values. Chroma value and total color change (Δ*E*) are calculated as given in Equations (5)–(6) [33].
(5)$$C={\left(a^{2}+b^{2}\right)}^{1/2}$$
(6)$$\Delta E=\;{\left[{\left(L-L_{0}\right)}^{2}\;+\;{\left(a-a_{0}\right)}^{2}\;+\;{\left(b-b_{0}\right)}^{2}\right]}^{1/2}$$
$L_{0},$
$a_{0},$ and $b_{0}$ show dried product brightness, redness, and yellowness, respectively. 

### 2.5. Solar Energy

The sun constant is the pre-atmosphere energy in direction of the radiation coming from the sun, perpendicular to the unit area at unit time, and the average distance between the sun-world is I_sc_ = 1367 W/m^2^. The intensity of the radiation on top of the collector was measured and recorded with a CEM DT-185 brand pyrometer during the drying process.

The available solar energy inside the solar dryer (Q) could be calculated in terms of the solar radiation that penetrated the cover and the net surface area of the dryer as given below:(7)$$Q=I\cdot A_{k}$$
(8)$$Q=\dot{m}C_{p}\Delta T$$
Q  is the thermal power generated in the collector, which is obtained using Equations (7) and (8) [34].

In Equation (8), C_p_ (=1005 J/kg °C) is the passing air-specific heat (J/kg °C), $\dot{m}\;$(kg/s) is the mass flow rate of the passing air (kg/s), and ∆T (°C) is the temperature difference between the input and output air from the collector.

The thermal efficiency (η) of the air solar collectors was calculated using Equation (9) [35].
(9)$$\eta =\frac{\dot{m}c_{p}(T_{\mathrm{out}}-T_{\mathrm{in})}}{{\mathrm{IA}}_{c}}$$

In this equation, $T_{\mathrm{out}}$ is the outlet temperature of the air from the collector, $T_{\mathrm{in}}$ is the inlet temperature of the air from the collector, I is the total solar radiation coming to the collector’s surface (W/m^2^), and $A_{c}$ is the collector’s surface area. The surface area of the air collector used in the experiments is 1.5136 m^2^. The rate of the mass flow was calculated by Equation (10);
(10)$$\dot{m}={\rho \mathrm{VA}}_{o}$$
where ρ is the air density, V shows the speed of the air flow, and $A_{o}$ shows the collector outlet’s cross-sectional area. 

### 2.6. Statistical Analyses

The results of specific energy consumption and drying time depend on slice thickness and microwave power level were analyzed using analysis of variance (one-way ANOVA). The Tukey’s test was applied to determine if the differences were significant. All statistical analyses were performed with SPSS (PASW Statistics 18, SPSS Ltd, Hong Kong, China). Differences with *p*-values less than 0.05 were considered significant.

## 3. Results and Discussion

### 3.1. Drying Behaviors

In this work, drying behaviors of black radish were examined with three different microwave power levels (0.7 kW, 1 kW, and 1.4 kW), different thicknesses (4 mm, 6 mm, and 8 mm) and different collector temperatures. As a result of these trials, when slice thicknesses increase, drying time increases. Collector exit temperature and collector center temperature are changing depending on environment temperature. Depending on this drying temperature, drying time is also changing. Inside of the tunnel, temperature is changing depending on both hot air coming from the tunnel and effects of solar rays (Figure 2). That is why it does not have a constant value. Black radishes with 4-, 6-, and 8-mm slice thicknesses were placed into the drying chamber at the same time during the experiments. The slice samples with three different thickness values were dried at a single drying power. It was applied in the same way for other power levels. Therefore, Figure 2 shows the common temperature values (temperature of collector center, collector output and drying) for 4-, 6-, and 8-mm slice thicknesses. The curves are decreased and increased depending on the weather conditions. A single graph was given to comparison of these temperature values for 0.7-, 1-, and 1.4-kW microwave power values. The average value of drying temperatures was measured as 32 ± 0.7 °C in all trials. The 0.7-kW power’s drying time lasted between 70 and 82 min, the 1-kW power’s drying time lasted between 60 and 65 min, and the 1.4-kW power’s drying time lasted between 43 and 63 min. For 1.4-kW power, the shortest drying time was determined. 

Microwave energy transforms to the thermal energy for higher relative moisture content food and product’s temperature begins to rise with some time for the first phase of the drying. Liquid which is inside the solid material is heated until boiling point quickly. When the vapor pressure of product’s moisture is higher than environment’s vapor pressure, the product begins to lose moisture. The second phase involves a rapid drying area. Thermal energy that is gained thanks to the microwave energy used to meet evaporation energy. When the boiling point of the inside liquid of the solid is not reached, it can be observed that temperature rises throughout the drying. The third phase is the decreasing drying area. As the energy required for evaporation is lower than thermal energy which is gained with microwave energy, the temperature of the moisture inside of the product can be higher than the boiling temperature [36].

As Boldor et al. (2005) [37] stated, when liquid evaporates because of the pressure difference, this constant drying period is the second phase. In the third phase, because of having less water, the material’s temperature can rise rapidly and burning can occur. In contrast to drying with hot air, in which the product’s final temperature is never higher than the air temperature, it is harder to control the product’s final temperature for microwave drying [8].

Many dryers exist in the field of dryer technologies. As per the literature, tomato drying lasted 6–8.5 h for solar energy drying [38], and carrot slice drying lasted 3.5–3.75 h for freeze drying, microwave energy drying, and a combination with hot air. Carrot slice drying lasted 9.5 h for freeze-drying alone [39]; quince was dried with convective drying for 10–50 h [40]; red chili pepper was dried with vacuum drying (50–75 °C and 0.05, 7, and 13 kPa) for 3–19.17 h [41]; squash-drying lasted 1.5–8 h with forced convection at 40–80 °C [42]; and apple slice drying lasted 5–25 min with microwave energy at 200–600 W [43]. This research results showed that our design drier was able to dry samples in a shorter time than other drying methods. This was a longer time than drying using only microwave energy. On the other hand, product quality obtained from this design is more advantageous.

### 3.2. Color Parameters

Color measurements were made for every experiment condition before and after drying for determining color changes in three repetitions. Color parameters of the dried products were compared with color parameters of the fresh products and total color losses were determined. As a result of the drying, the total color change parameter (Δ*E*) values for 0.7 kW power and for 4-mm, 6-mm, and 8-mm slice thicknesses were found to be 7.05, 12.90 and 5.95, respectively. The Δ*E* values for 1 kW power and for 4-mm, 6-mm, and 8-mm slice thicknesses were found as 19.14, 19.27, and 13.59, respectively. The Δ*E* values for 1.4 kW power and for 4-mm, 6-mm, and 8-mm slice thicknesses were found as 17.87, 22.76, and 34.61, respectively. When we consider the value changes of the brightness (ΔL) for 0.7 kW power and for 4-mm, 6-mm and 8-mm slice thicknesses, ΔL values were found as 4.90, 7.72, and 4.12, respectively. The ΔL values for 1-kW power and for 4-mm, 6-mm, and 8-mm slice thicknesses were found as 13.65, 12.97 and 9.51, respectively. The ΔL values for 1.4 kW power and for 4-mm, 6-mm, and 8-mm slice thicknesses were found to be 5.89, 7.84, and 1.38, respectively. As a result of the drying, when considering the brightness change parameter (ΔL) and total color change parameter (Δ*E*), the best results were obtained for 0.7-kW power and 8-mm slice thickness compared to other drying experiments. Appearance of the black radish after drying was given in Figure 3. Microwave drying of vegetables and fruits is attracting considerable attention because of the high mass transfer coefficients and the fact that it usually obtains better last product [44].

### 3.3. Energy Consumption Values

Appliances causing energy consumption involve the Teflon belt in the electrical engine, the magnetrons, and the fan, which are connected to the solar collector. Throughout the drying, energy consumption values of the experimental setup were measured for experiments performed with 4-mm, 6-mm, and 8-mm slice thicknesses for the 0.7-kW power level as 1.195, 1.245, and 1.361 kWh, for the 1-kW power level as 1.155, 1.213, and 1.252 kWh, and for the 1.4-kW power level as 0.980, 1.094, and 1.436 kWh, respectively. When microwave power increased, consumed energy also increased. Slice thicknesses were also compared and it was found that consumed energy amount also increased with increasing of slice thickness. However, for the drying experiment performed using the 1.4-kW power level, lower energy consumption values were obtained compared to the others. This is because the solar-heated air temperature is higher, and thus the drying time is shorter. Consequently, consumed energy was found to be lower. Effects of slice thickness on the specific energy consumption and drying time were found to be statistically insignificant (*p* > 0.05). On the other hand, effects of microwave power level on the specific energy consumption and drying time were found to be statistically significant (*p* < 0.05). Values of specific energy consumption and drying time determined for drying processes using the 0.7-, 1-, and 1.4-kW power levels were determined statistically in the different groups. Differences between three repetitions for specific energy consumption and drying time were found statistically insignificant (*p* > 0.05). Descriptive statistics for specific energy consumption and drying time determined for all drying conditions are shown in Table 2.

Instead of using microwave energy alone, using combined microwave conveyor and solar energy dryer provided time and energy savings significantly during works. In their works, Zarein et al. (2015) [43] heated cylindrical shape carrots, potatoes, and sweet potatoes to 50 °C and 80 °C by using microwaves, and they reported drying time was reduced with hot air use. For a continuous dryer with solar energy, the solar collector provided heat energy of 0.2 m s^−1^ air velocity for 40 °C at 1.908 kWh, for 45 °C at 2.043 kWh, and for 50 °C at 2.237 kWh [38]. In Table 3, specified energy consumption values are lower when compared to the other drying methods.

### 3.4. Modelling of The Drying Behaviours and Selection of The Suitable Model

Modelling of drying operation in microscopic dimensions is about material’s internal structure and usually independent of features of the equipment. Results of the coefficients of drying models *r*^2^, *e*_s_, and *χ*^2^ are given in Table 4, Table 5 and Table 6. When we consider for all slices with the values for *r*^2^ (0.950–0.998), for *e*_s_ (0.014–0.045), and for *χ*^2^ (0.0002–0.002), it was seen that the most suitable model was found to be the logistic model. Throughout the Figure 4, Figure 5 and Figure 6, comparisons of the drying curves depending on this model were given.

During the drying of the material with homogeneous and porous structure, physical changes and chemical reactions occur and these are affecting the moisture transfer mechanism inside the product. When it is considered that the food items’ structures are nonhomogeneous, issues in modelling become more understandable. Heat and mass transfer can be effective for the material’s physical and chemical properties, and as a cycle interactions are shown [45]. Combined use of the microwave energy with different drying methods increases the homogeneity of the temperature distribution inside the product, providing more control over the moisture transfer and also increasing the drying velocity [46]. 

### 3.5. Collector Efficiency

The instantaneous intensity of sunlight was measured during the experiments and the mean was found to be 1002 W/m^2^. The mass flow rate required to calculate the instantaneous thermal efficiency of the collector was found to be 0.04274 kg/s on average, calculated with the help of Equation (9). The instantaneous thermal efficiency of the solar collector was calculated to be 24.3%. Bulut et al. 2006 [35] found the efficiency of the collector they used in their study to be 53%. Aktaş et al. (2012) [38] calculated the efficiency of the collector they used in their study as 56.7%.

## 4. Conclusions

In this work, the drying operation for 0.7 kW power lasted between 70 and 82 min; for 1-kW power it lasted between 60 and 65 min; and for 1.4-kW power it lasted between 43 and 63 min depending on the slice thickness. The most suitable results for the required time of drying of the black radish slices until reaching 12 ± 0.5% (wet base) and for electrical energy consumption values were obtained using 1.4-kW microwave power, 0.245 m/min conveyor speed, and a 4-mm black radish slice thickness, at 0.980 kWh. Increasing microwave power for all slice thicknesses increased drying time and energy consumption values. Because of the high air temperature that was transferred to the drying tunnel, the trials that were performed for 1-kW power showed fast drying. That is why low consumption values were obtained. Also, specific energy consumption values were measured for 0.7-kW, 1-kW, and 1.4-kW microwave power levels at between 7.407 and 12.633 kWh/kg for 4 mm, between 4.894 and 10.449 kWh/kg for 6 mm, and between 3.671 and 11.863 kWh/kg for 8-mm slice thicknesses, respectively.

When color criteria were examined, the best results were obtained for 0.7-kW power drying operations with the 8-mm black radish slice compared to the other experiments, while taking in consideration (ΔL) and (Δ*E*) values. 

For predicting separable humidity rate, the “logistic” drying model was chosen as the most predictive drying kinetics model, with comparisons made depending on coefficients of the examined drying models, the highest *r*^2^ value based on microwave power, and slice thicknesses in the overall experimental conditions.

As a result, in the drying implementations with a microwave dryer with a conveyor it can be said that working with a low belt speed is possibly more beneficial for quality criteria, and in the conditions that the sun is effective, energy consumption values are lower. Also, it can be said that these methods are more feasible for drying of the fruits and vegetables where commercial issues, conservation of color, and the other quality parameters are important.

## Figures and Tables

**Figure 1 foods-08-00081-f001:**
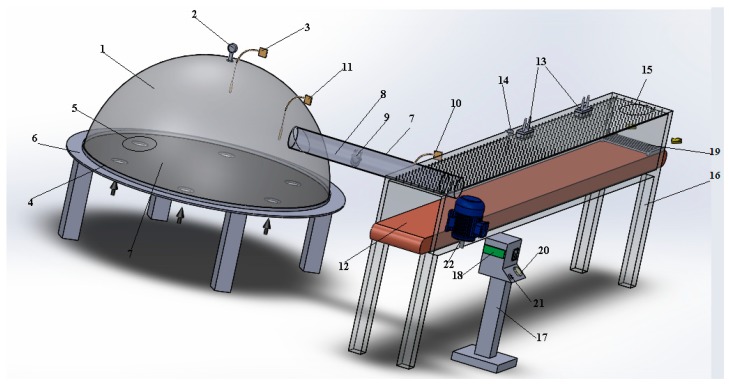
Overview of solar energy-assisted microwave conveyor dryer. (1) Collector room; (2) pyranometer; (3) thermocouples; (4) thermometer; (5) insulation; (6) platform; (7) collector; (8) channel; (9) fan; (10) anemometer; (11) thermometer; (12) conveyor; (13) magnetron (14) PT 100 thermometer; (15) drying room; (16) stand; (17) control panel; (18) energy meter; (19) tray; (20) power (on/off); (21) potentiometer; (22) electric motor.

**Figure 2 foods-08-00081-f002:**
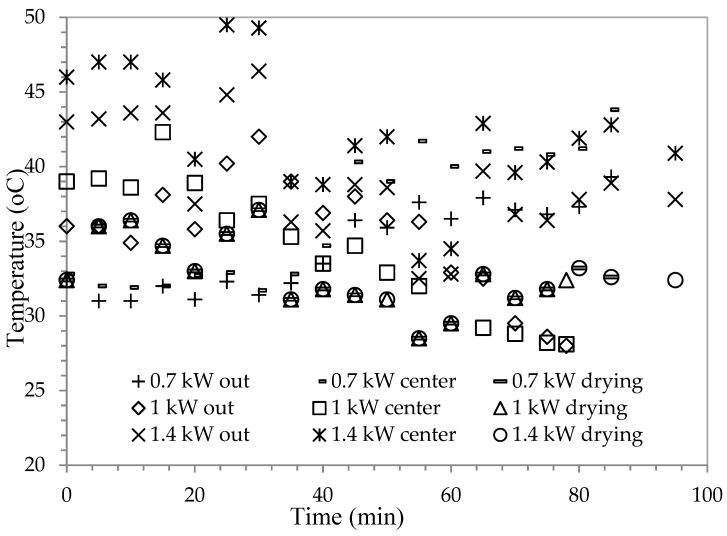
Temperature changes with time in the central point of the solar collector, solar collector exit, and drying entrance to the tunnel.

**Figure 3 foods-08-00081-f003:**
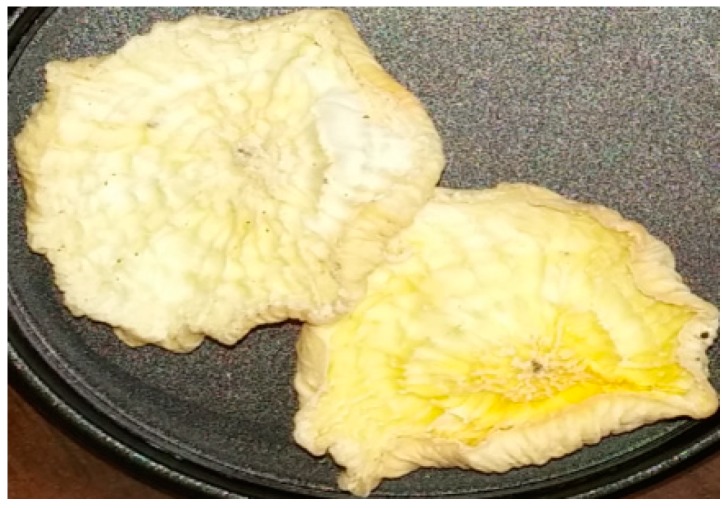
Photograph of the dried black radish slices.

**Figure 4 foods-08-00081-f004:**
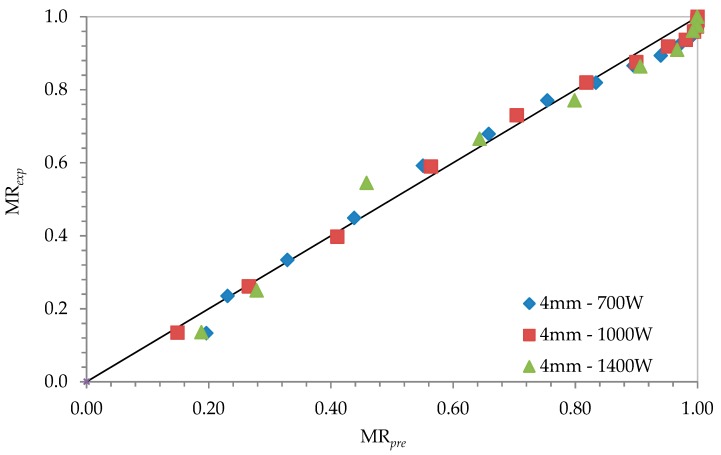
Drying curves depending on the logistic model with 4-mm slice thickness and 0.7-kW drying power.

**Figure 5 foods-08-00081-f005:**
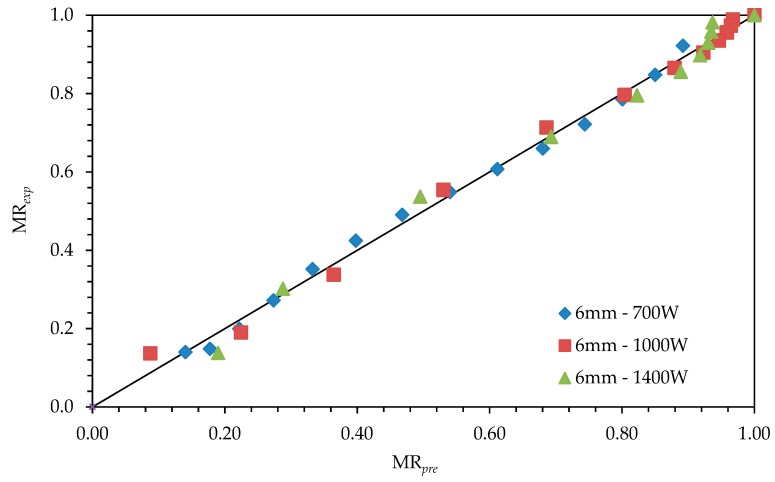
Drying curves depending on the logistic model with 6-mm slice thickness and 1-kW drying power.

**Figure 6 foods-08-00081-f006:**
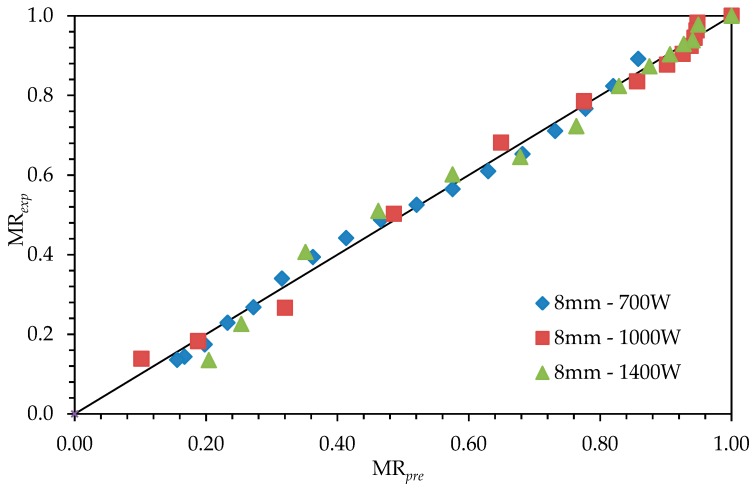
Drying curves depending on the Logistic Model with 8-mm slice thickness and 1.4-kW drying power.

**Table 1 foods-08-00081-t001:** Drying models [30].

Model	Model Equation
Newton	MR=exp(−kt)
Logarithmic	$\mathrm{MR}={\;a}_{0}+a\;\exp\left(-\mathrm{kt}\right)$
Logistic	$\mathrm{MR}={\;a}_{0}/\left(1+\mathrm{aexp}\left(\mathrm{kt}\right)\right)$
Midilli	$\mathrm{MR}=a\;\exp\left(-{\mathrm{kt}}^{n}\right)+\mathrm{bt}$
Two term	$\mathrm{MR}=a_{1}\exp\left(-k_{1}t\right)+a_{2}\exp\left(-k_{2}t\right)$
Verma	MR=aexp(−kt)+(1−a)exp(−gt)
Wang and Singh	$\mathrm{MR}=1+\mathrm{at}+{\mathrm{bt}}^{2}$

The coefficients in these models are determined by performing a nonlinear regression analysis. a_0_, a, a_1_ a_2_, b, n, k, k_1_, k_2_, and g are model variables; t represents time.

**Table 2 foods-08-00081-t002:** Descriptive statistics for specific energy consumption and drying time determined for all drying conditions.

Parameters	*n*	Minimum	Maximum	Mean	Standard Deviation
Specific energy consumption	27	3.55	12.79	8.1858	3.06355
Drying time	27	42.00	83.00	63.1111	11.15738
Valid *n* (listwise)	27				

**Table 3 foods-08-00081-t003:** Energy consumption values for black radish slices.

	Total Time (Min)	Energy Consumption (kWh)	Specific Energy Consumption (kWh/kg)
4 mm, 0.245 m/min 700 W	72	1.195	7.407
6 mm, 0.245 m/min 700 W	75	1.245	4.894
8 mm, 0.245 m/min 700 W	82	1.361	3.671
4 mm, 0.245 m/min 1000 W	60	1.155	9.779
6 mm, 0.245 m/min 1000 W	63	1.213	7.489
8 mm, 0.245 m/min 1000 W	65	1.252	5.451
4 mm, 0.245 m/min 1400 W	43	0.980	12.633
6 mm, 0.245 m/min 1400 W	48	1.094	11.863
8 mm, 0.245 m/min 1400 W	63	1.436	10.449

**Table 4 foods-08-00081-t004:** Analysis results of the models for the black radish slices with 4-mm thickness.

Models	M.P.	Coefficients	*r* ^2^	*e* _s_	*χ* ^2^
Newton	700 W	k: 0.09	0.816	0.020	0.140
	1000 W	k: 0.013	0.795	0.033	0.181
	1400 W	k: 0.019	0.822	0.032	0.178
Page	700 W	k: 1.9 × 10^−7^, n: 3.729	0.990	0.001	0.036
	1000 W	k: 7.2 × 10^−8^, n: 4.174	0.995	0.001	0.027
	1400 W	k: 1.5 × 10^−6^, n: 3.700	0.980	0.003	0.052
Henderson and Pabis	700 W	k: 0.015, a: 1.230	0.777	0.020	0.142
	1000 W	k: 0.019, a: 1.259	0.758	0.024	0.156
	1400 W	k: 0.028, a: 1.282	0.786	0.024	0.154
Wang and Singh	700 W	a: 0.003, b: −1.9 × 10^−4^	0.989	0.001	0.032
	1000 W	a: 0.005, b: −3.2 × 10^−4^	0.987	0.001	0.036
	1400 W	a: 0.005, b: −0.001	0.986	0.002	0.040
Two-term exponential	700 W	k: −0.048, a: −0.025	0.995	4.6 × 10^−4^	0.022
	1000 W	k: −0.060, a: −0.023	0.983	0.002	0.042
	1400 W	k: −0.080, a: −0.026	0.995	0.001	0.023
Logarithmic	700 W	k: 1.8 × 10^−4^, a: 55.978, a_0_: −54.885	0.877	0.015	0.121
	1000 W	k: 6.1 × 10^−5^, a: 251.366, a_0_: −250.149	0.864	0.015	0.122
	1400 W	k: 1.7 × 10^−4^, a: 111.659, a_0_: −110.49	0.888	0.015	0.123
Logistic	700 W	k: 0.104, a: 0.002, a_0_: 0.961	0.995	0.001	0.023
	1000 W	k: 0.139, a: 0.001, a_0_: 0.970	0.998	2.0 × 10^−4^	0.014
	1400 W	k: 0.180, a: 0.002, a_0_:0.950	0.950	0.002	0.045
Midilli	700 W	a: 1.211, k: 3.4 × 10^−18^, n: 9.068, b: −0.011	0.954	0.007	0.081
	1000 W	a: 1.251, k: 2.9 × 10^−13^, n: 6.721, b: −0.014	0.943	0.010	0.100
	1400 W	a: 1.218, k: 4.9 × 10^−22^, n: 12.577, b: −0.021	0.950	0.009	0.095
Two-term	700 W	a_1_: 61.051, k_1_: 0.041, a_2_: −60.310, k_2_: 0.042	0.915	0.009	0.097
	1000 W	a_1_: 44.853, k_1_: 0.052, a_2_: −44.261, k_2_: 0.054	0.908	0.012	0.110
	1400 W	a_1_: 16.849, k_1_: 0.070, a_2_: −16.293, k_2_: 0.080	0.914	0.014	0.117
Verma	700 W	a: −17.912, k: 0.038, g: 0.035	0.889	0.011	0.105
	1000 W	a: 1.259, k: 0.019, g: 28.477	0.758	0.027	0.165
	1400 W	a: 2.743, k: 0.019, g: 0.019	0.822	0.042	0.206

M.P. is Microwave Power.

**Table 5 foods-08-00081-t005:** Analysis results of the models for the black radish slices with 6-mm thickness.

Models	M.P.	Coefficients	*r* ^2^	*e* _s_	*χ* ^2^
Newton	700 W	k: 0.021	0.966	0.004	0.066
	1000 W	k: 0.013	0.811	0.035	0.186
	1400 W	k: 0.015	0.782	0.031	0.177
Page	700 W	k: 0.004, n : 1.429	0.984	0.001	0.036
	1000 W	k: 1.7 × 10^−7^, n : 3.970	0.989	0.001	0.038
	1400 W	k: 1.2 × 10^−7^, n: 4.217	0.981	0.003	0.051
Henderson and Pabis	700 W	k: 0.023, a: 1.106	0.957	0.003	0.058
	1000 W	k: 0.020, a: 1.274	0.772	0.025	0.159
	1400 W	k: 0.022, a: 1.249	0.748	0.025	0.157
Wang and Singh	700 W	a: −0.014, b: 1.8 × 10^−5^	0.996	3.0 × 10^−4^	0.017
	1000 W	a: 0.003, b: −2.8 × 10^−4^	0.967	0.004	0.060
	1400 W	a: 0.006, b: −4.6 × 10^−4^	0.976	0.003	0.050
Two-term exponential	700 W	k: 62.184, a: 3.3 × 10^−4^	0.966	0.005	0.069
	1000 W	k: −0.048, a: −0.044	0.946	0.006	0.080
	1400 W	k: −0.081, a: −0.017	0.999	6.4 × 10^−5^	0.008
Logarithmic	700 W	k: 4.9 × 10^−4^, a: 25.633, a_0_: −24.649	0.996	2.9 × 10^−4^	0.017
	1000 W	k: 0.002, a: 11.025, a_0_: −9.774	0.874	0.015	0.124
	1400 W	k: 2.7 × 10^−4^, a: 56.294, a_0_: −55.189	0.846	0.020	0.141
Logistic	700 W	k: 0.055, a: 0.141, a_0_: 1.057	0.994	4.6 × 10^−4^	0.022
	1000 W	k: 0.139, a: 0.002, a_0_: 0.971	0.994	0.001	0.028
	1400 W	k: 0.185, a: 0.001, a_0_: 0.938	0.988	0.001	0.037
Midilli	700 W	a: 0.980, k: 0.003, n: 1.052, b: −0.009	0.996	3.2 × 10^−4^	0.018
	1000 W	a: 1.252, k: 9.5 × 10^−6^, n: 2.029, b: −0.016	0.888	0.016	0.126
	1400 W	a: 1.207, k: 5.4 × 10^−27^, n: 15.227, b: −0.017	0.938	0.010	0.098
Two-term	700 W	a_1_: 10.585, k_1_: 0.044, a_2_: −9.709, k_2_: 0.049	0.987	0.001	0.035
	1000 W	a_1_: 20.007, k_1_: 0.051, a_2_: −19.349, k_2_: 0.057	0.916	0.012	0.110
	1400 W	a_1_: 13.581, k_1_: 0.058, a_2_: −12.951, k_2_: 0.068	0.885	0.015	0.123
Verma	700 W	a: 10.944, k: 0.041, g: 0.044	0.981	0.002	0.040
	1000 W	a: 4.093, k: −0.012, g: −0.017	0.960	0.005	0.070
	1400 W	a: 1.523, k: 0.015, g: 0.015	0.781	0.040	0.201

**Table 6 foods-08-00081-t006:** Analysis results of the models for the black radish slices with 8-mm thickness.

Models	M.P.	Coefficients	*r* ^2^	*e* _s_	*χ* ^2^
Newton	700 W	k: 0.019	0.969	0.002	0.047
	1000 W	k: 0.013	0.782	0.035	0.188
	1400 W	k: 0.014	0.858	0.020	0.142
Page	700 W	k: 0.009, n: 1.194	0.977	0.002	0.039
	1000 W	k: 4.8 × 10^−4^, n: 4.227	0.985	0.002	0.047
	1400 W	k: 1.2 × 10^−5^, n: 2.823	0.978	0.002	0.047
Henderson and Pabis	700 W	k: 0.019, a: 1.033	0.966	0.002	0.047
	1000 W	k: 0.018, a: 1.254	0.745	0.028	0.166
	1400 W	k: 0.019, a: 1.214	0.831	0.015	0.122
Wang and Singh	700 W	a: −0.014, b: 4.0 × 10^−5^	0.993	0.001	0.026
	1000 W	a: 0.004, b: −2.7 × 10^−4^	0.969	0.003	0.058
	1400 W	a: −2.5 × 10^−4^, b: 2.0 × 10^−4^	0.991	0.001	0.028
Two-term exponential	700 W	k: 29.314, a: 0.001	0.969	0.002	0.049
	1000 W	k: −0.052, a: −0.031	0.960	0.004	0.066
	1400 W	k: −0.045, a: −0.047	0.996	3.6 × 10^−4^	0.019
Logarithmic	700 W	k: 1.9 × 10^−4^, a: 51.475, a_0_: −50.549	0.997	2.1 × 10^−4^	0.015
	1000 W	k: 2.1 × 10^−4^, a: 71.238, a_0_: −70.024	0.855	0.017	0.130
	1400 W	k: 2.7 × 10^−4^, a: 50.138, a_0_: −48.993	0.921	0.007	0.086
Logistic	700 W	k: 0.041, a: 0.212, a_0_: 1.080	0.992	0.001	0.023
	1000 W	k: 0.145, a: 0.001, a_0_: 0.949	0.992	0.001	0.030
	1400 W	k: 0.095, a: 0.010, a_0_: 0.964	0.984	0.002	0.039
Midilli	700 W	a: 0.941, k: 0.008, n: 0.657, b: −0.008	0.998	1.6 × 10^−4^	0.013
	1000 W	a: 1.205, k: 3.8 × 10^−15^, n: 7.521, b: −0.014	0.903	0.014	0.118
	1400 W	a: 1.207, k: 2.3 × 10^−7^, n: 3.130, b: −0.014	0.945	0.008	0.087
Two-term	700 W	a_1_: 10.997, k_1_: 0.036, a_2_: −10.131, k_2_: 0.039	0.986	0.001	0.032
	1000 W	a_1_: 38.715, k_1_: 0.049, a_2_: −38.105, k_2_: 0.052	0.893	0.015	0.122
	1400 W	a_1_: 41.627, k_1_: 0.046, a_2_: −40.833, k_2_: 0.048	0.939	0.007	0.082
Verma	700 W	a: −2.600, k: 0.036, g: 0.029	0.978	0.002	0.039
	1000 W	a: 1.574 k: 0.013 g: 0.013	0.782	0.042	0.206
	1400 W	a: 2.944 k: 0.014 g: 0.014	0.858	0.024	0.156

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
