# Peer review of "Investigation of the Performance of a Hybrid Dryer Designed for the Food Industry"

_foods, 2019, doi:10.3390/foods8020081_

Reviewer 1 Report

Edit the English and grammar by an accredited editor

Also found some repetitions in some places, please rectify this (specially in the Introduction)

Justification for Microwave use in drying is not sufficient, use some more references to prove this

In material and methods carefully describe each steps used in material preparation, how did you minimise errors in sample sizes, also how did you control the temperature of drying so precisely with out any error?

Include an error estimation?

A conveyor was mentioned but couldnot find it in the Figure 1, I suspect it is named as Band. Use same word to avoid confusion to the readers

Where ids the colour evaluations, this should be included in the revised version of the manuscript

Author Response

Dear Reviewer,

First of all, I want to thank you for your kind suggestions and comments for my article. The publication of the article is very important for my academic advancement. I took your suggestions into consideration and did the necessary corrections in the article. I hope that all of them are enough for you. The corrections are given below.

Point 1: Edit the English and grammar by an accredited editor

Response 1: Thank you for your suggestion. The English grammar confirmation of the article was corrected by an expert.

Point 2: Also found some repetitions in some places, please rectify this (specially in the Introduction)

Response 2: Repeated sentences in the introduction section have been deleted as you suggested.

Point 3: Justification for Microwave use in drying is not sufficient, use some more references to prove this

Response 3: References 28 and 29 was added. Thank you for your suggestion.

Point 4: In material and methods carefully describe each steps used in material preparation, how did you minimise errors in sample sizes, also how did you control the temperature of drying so precisely with out any error?

Response 4: Each step used in material preparation has been added as “The black radishes were peeled and sliced at 4 ± 0.3 mm, 6 ± 0.2 mm and 8 ± 0.2 mm as parallel to the axis by knife for the experiments. Different samples were used for each experiment” in 2.1 section. Change of temperature in the channel and collector was shown in Figure 2. The control of drying temperature is difficult due to fluctuations in temperature. Drying processes were carried out according to the changing temperature.

Point 5: Include an error estimation?

Response 5: We agree with your suggestion. But, we did not make an error estimation.

Point 6: A conveyor was mentioned but could not find it in the Figure 1, I suspect it is named as Band. Use same word to avoid confusion to the readers

Response 6: It has been corrected as conveyor. Thank you for your suggestion.

Point 7: Where is the colour evaluations, this should be included in the revised version of the manuscript.

Response 7: Total colour change and brightness change have been shown in 3.2 section. But, we didn't need to show the redness/greenness and yellowness/blueness separately. We hope it is enough. Thank you for your suggestion.

Reviewer 2 Report

The topic of using solar-heated air for drying is very interesting. The research pilot plant also has great research potential.
However, a weak way of presenting both the methodology and the results obtained does not allow the substantive evaluation of the obtained results.
Therefore, it is necessary to correct the methodology, present the results and discuss them.
Detailed comments are provided in the attached text

Author Response

Dear Reviewer,

First of all, I would like to thank you for your kind suggestions and comments for my article. The publication of this article is very important for my academic advancement. I took your suggestions into consideration and did the necessary corrections on the article. I hope that all of them are enough for you. The corrections are given below.

Point 1: No conduction, heat is generated inside the material.

Response 1: The sentence defined in Line 48 was removed as suggested.

Point 2: If it is known dependencies, what was the purpose of the research?

Response 2: The subject I mentioned in line 77-79 is an assumption. It is stated that it is advantageous according to the conventional methods that the studies conducted with solar and microwave drying in the literature. We aimed to improve the drying system and the dried sample combining these systems.

Point 3: The mass of the dried material is necessary. How was the dried material weighed? No drying parameters in the methodology (microwave power, velocity of flowing air, belt speed) the description of the dryer shows that it works continuously. Therefore, the material reaches the microwave zone only after some time and time in the microwave field is limited. Detailed information is necessary.

Response 3: Detailed information was given in the section 2.2 as you suggested. Thanks for your suggestion.

Point 4: If the thickness grows twice, why does the mass change only between 9 - 12 g.

Response 4: Not only the thickness of the samples but also the diameters are different. This makes variable the weights of the product.

Point 5: SI system should be used the unit of the temperature difference is Kelvin.

Response 5: It has been corrected as “K”.

Point 6: The results calculated by equations 7-10 are not presented in the paper.

Response 6: Efficiency was given in section 3.5.

Point 7: how the influence of 1 independent variable was determined, if in each experiment there were 2 variables: drying air temperature and microwave power or thickness of patches the experience plan is necessary.

Response 7: The variable numbers are more as you specified. The change in the collector and the drying temperature and the change in microwave power affect drying. It was about the thickness that we mentioned.

Point 8: check whether these values differ statistically.

Response 8: These values are the data I obtained as a result of my experiments.

Point 9: why this discussion if the material temperature was not measured?

Response 9: What I want to indicate here is, this is a general situation that occurs in microwave drying. I commented this way the results that I obtained.

Point 10: quoted values in no way relate to the experience, so you cannot compare the results with those values.

Response 10: In general, I compared my system with the drying systems in the literature. Our aim was to show that our system was advantageous to different products and dryers.

Point 11: The graph is not legible. In the text, no discussion of the results presented in the chart. Give the average of three replicates and the standard deviation

Response 11: we wanted to show all the temperature values in a single graph. This is already average temperature values. There is no discussion in the text because there is no such information. Thank you for your suggestion.

Point 12: there were different drying times, so values should be given with mean values and standard deviations statistically significant differences should be checked.

Response 12: I agree with you. Thank you for your suggestion.

Point 13: there was no experience without heating the air, so it can not be compared.

Response 13: Comparisons are related to hot air dryer. Comparison can be made. I disagree with you.

Point 14: to assess energy consumption, they should be expressed in kWh per kg of evaporated water.

Response 14: You mention about specific energy. We mentioned the energy consumption of the system. Some work is done as the way you say. My goal was to determine the consumption of the system.

Point 15: there is no equation 11 in the methodology.

Response 15: It has been corrected as “Eq. 9”.

Point 16: The statement is not justified in the presented results

Response 16: This is a suggestion for further studies.

Round  2

Reviewer 1 Report

Check the manuscript again carefully to remove any missing points

Author Response

Dear Reviewer,

First of all, I want to thank you for your kind suggestions and comments for my article. 

Best regards

Reviewer 2 Report

Dear authors,

I believe that some correction are still necessary.

Comments to the author's response:

Point 8: Check whether these values differ statistically.

Response 8: These values are the data I obtained as a result of my experiments.

reviewer's comment

(286 do 288) “0.7 kW power’s drying time lasted between 70-82 minutes, 1 kW power’s drying time lasted between 60-65 minutes and 1.4 kW power’s drying time lasted between 43-63 minutes”. You compare the ranges of values. To determine if the average values differ, the hypothesis that these values are equal, should be verified. Only the rejection of the hypothesis allows to state that drying time 43-63 is shorter than 60-65 minutes

Point 11: The graph is not legible. In the text, no discussion of the results presented in the chart. Give the average of three replicates and the standard deviation

Response 11: we wanted to show all the temperature values in a single graph. This is already average  temperature  values.  There  is  no  discussion  in  the  text  because  there  is  no  such information. Thank you for your suggestion.

reviewer's comment

For the drying process the most important is the temperature of the inlet air to the drying chamber. And this temperature should be particularly analyzed. Showing the standard deviation allows you to assess the variability of conditions in subsequent drying.

Point 12: there were different drying times, so values should be given with mean values and

standard deviations statistically significant differences should be checked.

Response 12: I agree with you. Thank you for your suggestion.

reviewer's comment

“for 0.7 kW power level as 1.195, 1.245 and 1.361 kWh, for 1 kW  power level as 1.155, 1.213 and 1.252 kWh and for 1.4 kW power level as 0.980, 1.094 and 1.436 kWh respectively” The author did not submit the results to the statistical evaluation. Subjecting these values to statistical analysis (the reviewer did it), it shows that statistically they are the same. Thus, the conclusions are incorrect

Point  14:  to asses  energy consumption, they should be  expressed  in  kWh  per  kg of evaporated water.

Response 14: You mention about specific energy. We mentioned the energy consumption of the  system.  Some  work  is  done  as  the  way  you  say.  My  goal  was  to  determine  the consumption of the system.

reviewer's comment

The total energy consumption depends on the installed power and working time - so there is no need to measure it. To compare the energy consumption in a variety of conditions you need to determine the specific energy consumption

I have no comments to correction the remaining points.

Author Response

Dear Reviewer,

First of all, I want to thank you for your kind suggestions and comments for my article. I took your suggestions into consideration and did the necessary corrections in the article. I hope that all of them are enough for you. The corrections are attached.

Point 8: check whether these values differ statistically.

Response 8: Statistical analysis you suggested was performed. Also, statistical information was shown in Table 3 and section 3.3.

Table 3. Descriptive statistics  for specific energy consumption and drying   time determined for all drying conditions

N

Minimum

Maximum

Mean

Std. Deviation

Specific energy consumption

27

3,55

12,79

8,1858

3,06355

Drying time

27

42,00

83,00

63,1111

11,15738

Valid N (listwise)

27

Point 11: The graph is not legible. In the text, no discussion of the results presented in the chart. Give the average of three replicates and the standard deviation.

Response 11: The drying temperature varied according to the weather conditions during the experiments. This change is shown in Figure 2. The drying temperature was measured as 32 ± 0.7 oC in all trials and was stated in the text (in section 3.1).

Point 12: there were different drying times, so values should be given with mean values and standard deviations statistically significant differences should be checked.

Response 12: Statistical analysis was performed as you suggested. Also, statistical information was shown in Table 3 and section 3.3.

Point 14: to assess energy consumption, they should be expressed in kWh per kg of evaporated water.

Response 14: Specific energy consumption (kWh / kg) values were added in Table 2 and in the conclusion section as you suggested.
